# Kinesiology Students’ Perception Regarding Exercise Oncology: A Cross-Sectional Study

**DOI:** 10.3390/ijerph19137724

**Published:** 2022-06-23

**Authors:** Alice Avancini, Carlo Ferri Marini, Isabella Sperduti, Valentina Natalucci, Anita Borsati, Sara Pilotto, Claudia Cerulli, Elena Barbieri, Francesco Lucertini, Massimo Lanza, Attilio Parisi, Elisa Grazioli, Andrea Di Blasio

**Affiliations:** 1Department of Oncology, University of Verona Hospital Trust, 37134 Verona, Italy; alice.avancini@univr.it (A.A.); sara.pilotto@univr.it (S.P.); 2Department of Neurosciences, Biomedicine and Movement Sciences, University of Verona, 37129 Verona, Italy; borsatianita8@gmail.com; 3Department of Biomolecular Sciences, University of Urbino Carlo Bo, 61029 Urbino, Italy; carlo.ferrimarini@uniurb.it (C.F.M.); valentina.natalucci@uniurb.it (V.N.); elena.barbieri@uniurb.it (E.B.); francesco.lucertini@uniurb.it (F.L.); 4Biostatistical Unit—Clinical Trials Center IRCCS Istituto Nazionale Tumori Regina Elena, U.O. di Biostatistica e Bioinformatica, 00144 Rome, Italy; isabella.sperduti@ifo.it; 5Department of Movement, Human and Health Sciences, University of Rome “Foro Italico”, 00135 Rome, Italy; claudia.cerulli@uniroma4.it (C.C.); attilio.parisi@uniroma4.it (A.P.); elisa.grazioli@uniroma4.it (E.G.); 6Department of Experimental and Clinical Medicine, “Magna Graecia” University, 88100 Catanzaro, Italy; 7Department of Medicine and Aging Sciences, “G. D’Annunzio” University of Chieti-Pescara, 66100 Chieti, Italy; andiblasio@gmail.com

**Keywords:** cancer, exercise, exercise specialists, kinesiologist, exercise oncology, university

## Abstract

Delivering physical activity in cancer care requires knowledge, competence, and specific skills to adapt the exercise program to the patients’ specific needs. Kinesiology students could be one of the main stakeholders involved in the promotion of physical activity. This study aims to investigate the knowledge, perception, and competence about exercise in patients with oncological disease in a sample of students attending the Sports Science University. A total of 854 students (13% response rate) from four Italian universities completed the online survey between May and June 2021. About half of the study participants identified the correct amount of aerobic (44%) and strength (54%) activities proposed by the American College of Sports Medicine for patients with cancer. Almost all the students recognized the importance of physical activity in cancer prevention (96%), in the management of cancer before surgery (96%), during anticancer treatments (84%), and after therapies completion (98%). On the contrary, they reported a lack of university courses dedicated to cancer diseases, psychological implications, and prescription of physical activity in all types of cancer prevention. Overall, few students felt qualified in delivered counseling about physical activity and individual or group-based exercise programs in patients with cancer. Logistic regression revealed that the students attending the Master’s Degree in Preventive and Adapted Physical Activity were more likely to have knowledge and competence than other students. The present study suggests that kinesiology universities should increase the classes and internships about exercise oncology to train experts with specific skills who are able to adequately support patients in their lifestyle modification.

## 1. Introduction

Most recent worldwide estimates report that approximately a quarter (27.5%) of adults are insufficiently active [1]. Physical inactivity is an established global pandemic that causes an elevated economic burden [2], but it is also a recognized risk factor for premature mortality and several non-communicable diseases, including cancer [3].

Cancer disease represents one of the leading causes of mortality worldwide, and its incidence and prevalence are expected to increase [4]. Although cancer causes may be genetic, modifiable risk factors, such as lifestyle and environmental exposure, also play an important role in the cancer trajectory. Among these, physical activity (PA) has emerged as a promising strategy for preventing and managing cancer [5]. In this sense, a series of epidemiologic evidence indicates that a scarce PA level is associated with the occurrence of many cancers (such as breast, colon, endometrium, and stomach) [5]. On the other hand, PA and exercise may also improve survival in patients with cancer (especially in the breast, colon, and prostate) [5] and manage some treatment-related side effects, like fatigue, nausea, and peripheral neuropathy [6].

Different national and international organizations recommend that patients with cancer should engage in regular exercise, comprised of aerobic and strength activities [6,7,8]. Despite the recognized benefits, a large majority of patients with cancer are insufficiently active [9]. Patients with malignancies may encounter different barriers hampering their participation in an exercise program, including lack of confidence, lack of time and information, cancer-related side effects, advanced disease symptoms, and medical procedures [10].

Nevertheless, translating research into practice remains a challenge. On this matter, a series of stakeholders, such as oncology clinicians, healthcare professionals, and kinesiologists, are involved in supporting exercise in people living with and beyond cancer [11], and it is crucial that all the experts would be proactive in its promotion. Multiple studies have investigated the perspectives of healthcare providers about *exercise oncology.* Although clinicians recognize the importance of exercise in the cancer context, only a limited proportion of them recommend to their patients to engage in PA, and even less refer them to an exercise program [12,13]. Lack of time during visits is the most common obstacle to promoting exercise in patients with cancer [14]. Moreover, clinicians identified the lack of trained exercise specialists and limited availability of specific programs as barriers seriously hindering exercise promotion [14].

Delivering exercise in the cancer population requires knowledge, competence, and confidence skills by specialists. Few training pathways (e.g., CanRehab, ACSM/ACS Cancer Exercise Trainer Certification) are available for preparing instructors to safely program exercise in the cancer population, and in some countries, like Italy, these are completely missing [12]. On this point, the education system, like Sports Science (Kinesiology) University, may play a crucial role in training exercise experts able to counsel, program, and lead exercise for patients with cancer. Italy’s sports science education system is characterized by two levels of degree: bachelor and master. The Bachelor’s Degree in Sport and Exercise Science is a three-year course offering general preparation for PA, including exercise for education, sports, and recovery of physical fitness. After that, three in-depth master’s degrees are currently available: (1) the Master’s Degree in Sports Science and Physical Performance is focused on sporting and educational activities; (2) the Master’s Degree in Sports Management is characterized by marketing and managerial courses related to PA; and (3) the Master’s Degree in Preventive and Adapted Exercise Science aims to train students able to adapt exercise in people with chronic diseases. According to the Italian law framework, individuals with a Bachelor’s Degree in Sports and Education Science are allowed to program and deliver exercise to patients with cancer.

However, to our knowledge, data are currently missing on students attending Sports Science University about their perspectives on *exercise-oncology*. To fill this gap and explore the current education program, the present study aimed to investigate the knowledge, perception, and expectation of students attending Sports Science University in Italy about the theme of *exercise-oncology*.

## 2. Materials and Methods

### 2.1. Study Design and Participants

A multicenter, online, cross-sectional study was proposed for 6658 students.

Student eligibility criteria were: (i) age ≥ 18 years, (ii) attending a degree in Sports Science, and (iii) adequate Italian language proficiency to answer the questionnaire. Potential participants were recruited from the University of Verona, Urbino, Rome (“Foro Italico”), and Chieti-Pescara from May to June 2021. Each university invited its students to complete an online survey (Google^®^ Form) through a direct email. Two reminders were sent every two weeks later to increase adherence. A cover letter describing the purpose of the study and the assurance of confidentiality and anonymity was presented before filling out the questionnaire.

The study was designed and conducted following the Declaration of Helsinki and approved by the local bioethical committee of the University of Rome’s “Foro Italico” (Prot. N. CAR 88/2021). It was carried out following the Strengthening the Reporting Observational studies in Epidemiology (STROBE) guidelines [15].

### 2.2. Questionnaire Description

The questionnaire was designed to assess sports science students’ knowledge, perception, and expectation about *exercise oncology*. The survey, developed after a literature review [16,17,18,19], comprised 33 items, divided into five sections: *(**a) socio-demographic and academic characteristics,* (*b) knowledge about exercise guidelines in cancer, (c) benefits of exercise in cancer, (d) presence of dedicated course(s) about exercise oncology at the university,* and *(e) confidence to deliver counseling and exercise programs to patients with cancer*.

Socio-demographic and academic information included: age, gender, weight, height, having or have had a cancer diagnosis, having or have had a family member with a cancer diagnosis, the attended university, academic degree, year of attendance, and participation in a non-academic course on exercise oncology.

Knowledge about exercise guidelines for patients with cancer was tested with two closed questions with multiple answers. The beneficial role of PA and exercise in cancer, as prevention and adjunctive therapy before surgery, during cancer treatments, and after therapies conclusion, was investigated through 4 items using a 4-point Likert scale ranging from *absolutely no* to *absolutely yes*. Six items were dedicated to exploring the presence of courses and/or specific lessons about cancer pathophysiology and treatment, psychological implications, and prescription of exercise in different cancer settings (i.e., before surgery, during cancer treatments, and after therapies conclusion), utilizing the abovementioned Likert scale. Finally, the perception of competence in exercise counseling, programming, and leading, was tested with 12 items using the 4-point Likert scale. A copy of the utilized questionnaire was available as Appendix A.

### 2.3. Statistical Analysis

Descriptive statistics, presented as frequency counts and percentages, were calculated for demographic and academic variables, as well as used to determine the knowledge of guidelines, the role of PA in cancer, the presence of specific teachings at the university, and the perception of competence to deliver exercise in the cancer population. A multivariate logistic regression model was used to examine the potential association between socio-demographic and academic variables with knowledge and competence in exercise oncology. The odds ratio (OR) and the 95% confidence interval (95% CI) were estimated for all variables of interest (age, gender, body mass index, having or have had a cancer diagnosis, having or have had a family member with a cancer diagnosis, university, the attending academic course). A multivariate proportional hazard model was developed using stepwise regression (forward selection, enter limit and remove the limit, *p* = 0.10 and *p* = 0.15, respectively) to identify independent predictors of outcome, considering the variables significant at univariate analysis (*p* ≤ 0.05).

The minimum sample required was 200 students. The expected sample allowed estimates of binary variables, e.g., percentages of students knowing the exercise guidelines (*p*) vs. those who do not recognize it (*p* = 1 − *p*), with a standard error of 0.035 and a confidence interval between 0.43 and 0.57, assuming the most unfavorable proportion equal to 0.5 (*p* = 0.5) and alpha 5%.

The SPSS (version 21.0; SPSS, Inc., Chicago, IL, USA), a licensed statistical program, was used for all analyses.

## 3. Results

### 3.1. Socio-Demographic and Academic Characteristics

A total of 6658 students were approached by email. After two reminders, 854 participants completed the survey, for a response rate of 13%. Demographic and academic information of the sample is presented in Table 1. The mean age of the participants was 24 years (SD = 5.7), 52% were male, 98% of them never had a cancer diagnosis, and 63% had family members with a story of oncological disease. Fifty-eight percent attended a Bachelor’s Degree in Sports Science, whereas 33% joined a Master’s Degree in Preventive and Adapted Exercise Science.

### 3.2. Knowledge of Exercise Guidelines for Patients with Cancer

Figure 1 displays the knowledge of the American College of Sports Medicine’s guidelines for patients with cancer. Overall, 44% and 54% of participants identified the correct amount for aerobic and strength activity, respectively. Multivariable logistic regression revealed that subjects having a family member with a cancer diagnosis were more likely to recognize the correct amount of aerobic activity (OR = 1.36 95% CI = 1.02 to 1.80) compared to those who did not have them. Female participants (OR = 1.46 95% CI = 1.11 to 1.92) and students who attended a Master’s Degree in Preventive and Adapted Exercise Science (OR = 2.18 95% CI = 1.61 to 2.95) had greater knowledge about the suggested frequencies to perform strength training compared to subjects that were male and joined the bachelor’s degree course.

### 3.3. Benefits of Exercise in Cancer

Almost all the students (96%) recognized the importance (i.e., “yes, rather than no” and “absolutely yes”) of PA in cancer prevention (Table 2). Focusing on patients with cancer, 96% and 98% of participants believed that exercise is important and effective both before cancer surgery and after treatment completion, whereas this percentage slightly drops to 84% when considered exercise during anticancer treatments. Students attending the Master’s Degree in Preventive and Adapted Exercise Science were more likely to consider the importance of exercise during anticancer treatments compared to students who joined the Bachelor’s Degree (OR = 1.76 95% CI = 1.13 to 2.73). The Appendix A displays students’ knowledge and perception according to the academic degree. The Appendix A reports the logistic regression model.

### 3.4. University Courses Dedicated to Exercise Oncology

More than half of students (62%) reported that in their academic degrees, there were courses explaining the prescription of PA as prevention of chronic non-communicable disease (Table 2). Overall, 40% and 30% said that there were lessons dedicated to the physiopathology and treatments of oncological disease and the psychological implication of cancer, respectively. A total of 39% of the students reported the presence of courses in their degree programs explaining the prescription of PA and exercise for patients with cancer before surgery, 36% during treatments, and 44% after therapy conclusion. These percentages slightly diminished when we considered a Bachelor’s Degree or Master’s Degree in Sports Science and Physical Performance or Sports Management, whereas they increased in students attending a Master’s Degree in Preventive and Adapted Exercise Science. The logistic regression shown in Appendix A revealed that the Master’s Degree in Preventive and Adapted Exercise Science was significantly related to the presence of oncology and exercise oncology courses.

### 3.5. Confidence in Delivering Counseling, and Exercise Programs to Patients with Cancer

Overall, the percentage of students that felt qualified to deliver counseling, prescribe exercise, and lead group-based or individual-based exercise training ranged from 22% to 34% in the different cancer settings (i.e., before surgery, during anticancer treatments, and after therapies conclusion) (Table 2). Students reported similar responses through academic degrees (Appendix A). Master’s Degree students in Preventive and Adapted Exercise Science reported having more knowledge to deliver counseling, prescribe exercise, and lead group-based or individual-based exercise training compared to participants joining bachelor’s degrees or other master’s degrees. These findings were supported by the logistic regression models (Appendix A).

Moreover, the knowledge of the aerobic exercise guidelines for patients with cancer was associated with more confidence in delivering PA counseling to patients with cancer before surgery (OR = 1.43 95% CI = 1.05 to 1.95) and during anticancer treatments (OR = 1.40 95% CI = 1.00 to 1.94). Similar results were obtained for the exercise prescription [before surgery (OR = 1.45 95% CI = 1.06 to 1.99) and during anticancer treatments (OR = 1.70 95% CI = 1.20 to 2.41)], the leading of group-based exercise training [before surgery (OR = 1.56 95% CI = 1.14 to 2.14), during anticancer treatments (OR = 1.71 95% CI = 1.20 to 2.41), and after therapies conclusion (OR = 1.45 95% CI = 1.08 to 1.96)], and the leading of individual-based exercise training [before surgery (OR = 1.45 95% CI = 1.07 to 1.96), during anticancer treatments (OR = 1.62 95% CI = 1.16 to 2.26), and after therapies conclusion (OR = 1.47 95% CI = 1.09 to 1.98)].

## 4. Discussion

To our knowledge, this is the first investigation exploring the knowledge, perception, and confidence toward exercise oncology of students attending four Italian sports science universities. This study highlights that approximately 50% of kinesiologists’ students know the exercise guidelines for patients with cancer, and more than 80% recognize the importance of exercise in the cancer continuum, but the majority report a lack of knowledge and expertise for counseling, prescription, and leading of exercise in patients with cancer.

About 44% and 54% of the students participating in the survey recognized the correct amount of aerobic and strength exercise recommended for patients with cancer. Prior studies on oncologists and healthcare providers reported less awareness about exercise guidelines than observed in the present study [20,21]. For instance, a recent investigation on oncology providers working in the lung cancer setting found that 39% recommended their patients perform 90 min per week of aerobic exercise, and 27% instructed them to include strength activities at least twice per week [22]. At first sight, our finding may be considered discouraging, especially considering the background of the participants, and it underlines the necessity to increase the classes dedicated to exercise oncology. Indeed, the knowledge of exercise guidelines specifically designed for cancer patients is essential to deliver a safe and feasible program and obtain the benefits from exercise [6].

Nevertheless, this result may be partially justifiable considering the limited presence of classes dedicated to the prescription of PA and exercise for patients with cancer. However, knowledge of the correct amount of exercise for patients with cancer was numerically higher than the presence of dedicated classes at their universities, thus suggesting that intrinsic motivation and a proactive approach to the academic path may be crucial. In this sense, logistic regression revealed that participants having a family member with a cancer diagnosis and being women are more likely to know the correct amount of physical exercise in the field of exercise oncology. Whether it is simple to understand the role of intrinsic motivation connected with having a family member with a cancer diagnosis, it is possible to speculate that the female gender is also linked with intrinsic motivation. As literature will testify the correlation among all grades of prevention of female cancers and physical exercise [5,6,23], and this is a *hot* sensitive topic for public opinion, being a woman gave the intrinsic motivation to prevent a disease that a female student feels nearer [24].

Almost all the students recognized the importance of PA and exercise as strategies to prevent cancer and as tools to increase the quality of life and obtain a range of benefits for patients with cancer, before surgery, during, and after medical treatments. The importance of exercise in oncology is rapidly growing, both among patients and healthcare providers [10,22]. This is an important starting point for the professionals involved in the health alliance, as the belief regarding the perceived benefits enhances the positive behaviors addressed to reducing the disease threat. Indeed, different psychological models, such as the Health Belief Model, include the perceived benefits as a predictor of action [25]. Thus, it is possible to speculate that beliefs about the benefits related to exercise might help the students to promote an active lifestyle in patients with cancer. The fact that, notwithstanding the low number of dedicated classes, almost all the students recognized the importance of PA and exercise in all grades of cancer prevention is positive information about the quality of the basic knowledge provided by the considered universities. Therefore, it is not possible to completely exclude the fact that one part of students could have considered PA and exercise useful against cancer simply because “*it is generally known that they are useful against diseases*”.

We found that a low percentage of students felt competent in delivering counseling and exercise programs to patients with cancer. Moreover, just about half of the students felt qualified to counsel or prescribe exercise and deliver group-based or individual-based exercise training in different cancer settings. These results are comparable with previous research on family medicine residents [26], and they could be related to the lack of knowledge regarding cancer and its therapy as well as the lack of teachings explaining exercise prescription in patients with cancer. Moreover, students attending a Master’s Degree in Preventive and Adapted Exercise Science reported more frequently the presence of oncology and exercise oncology courses compared to other degrees, and this can reflect on confidence in promoting exercise. Indeed, the attendance of a Master’s Degree in Preventive and Adapted Exercise Science was significantly associated with more confidence to counsel, deliver, and lead an exercise program for patients with cancer. In this sense, it is fundamental to increase teachings and curricular internships to increase students’ competencies in this field.

Our study presents some limitations. First, the response rate was low, only 13%, suggesting a general interest in the topic and making our result little generalizable. Secondly, even if the study actively involved 854 university students, the main limitation is related to the involvement of only a limited number of Italian sports science universities. Therefore, the observed results could not be fully representative of the Italian academic situation, also considering the existence of an active national research group that is highly contributing to teaching and research activities about the role of PA and exercise in all-grade prevention of cancer. Third, we did not ask about workplace learning or placements in our questionnaire, which may provide important information about students’ competence and knowledge. Future studies should consider these aspects to offer a more comprehensive picture of this topic.

## 5. Conclusions

In conclusion, our study provides an important starting point to improve the training of kinesiologists to be able to stimulate healthcare providers to increase physical exercise prescription and to support patients with oncological diseases with PA counseling, appropriate exercise prescription, and conduction. We found that kinesiology students are aware of the benefits of exercise in the cancer setting. Nevertheless, only a small percentage feel competent to provide PA counseling and program exercise in patients with cancer. Since kinesiologists are one of the experts that may deliver exercise to the cancer population, sports science universities should implement their academic offers, including dedicated internships, about the all-grade prevention of cancer.

## Figures and Tables

**Figure 1 ijerph-19-07724-f001:**
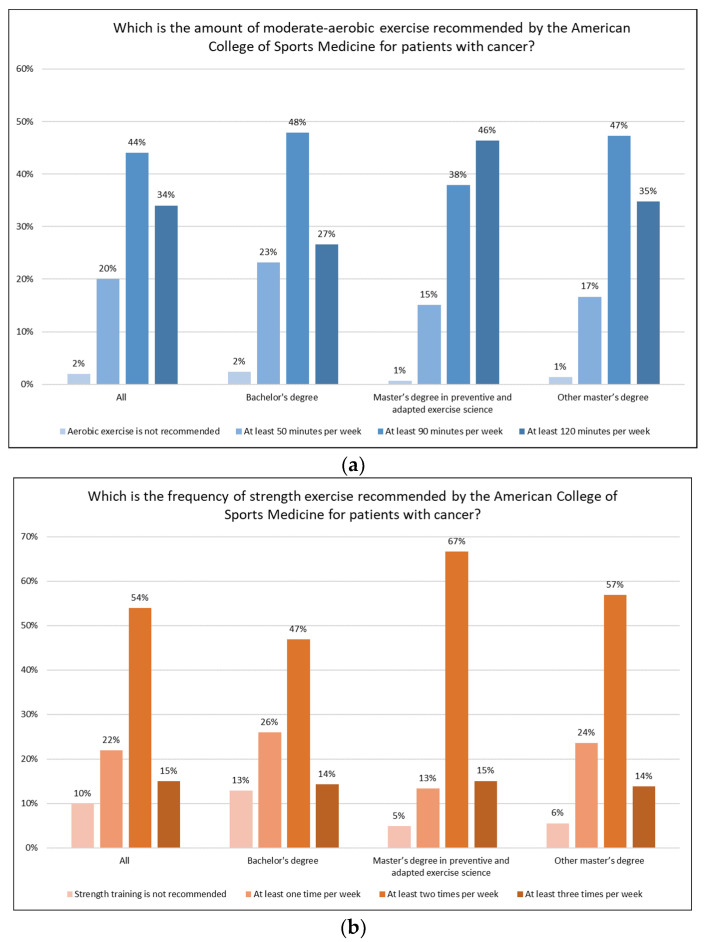
Knowledge of the American College of Sports Medicine’s guidelines for patients with cancer. (**a**) Knowledge of aerobic exercise guidelines; (**b**) knowledge of strength exercise guidelines.

**Table 1 ijerph-19-07724-t001:** Characteristics of the study’s participants.

Characteristic	Number	Percentage
*Sex*		
Female	411	48
Male	443	52
*Having or have had a cancer diagnosis*		
Yes	15	2
No	839	98
*Having a family member with a cancer diagnosis*		
Yes	538	63
No	316	37
*University*		
Verona	138	16
Urbino	179	21
Roma	252	30
Chieti-Pescara	285	33
*Academic course*		
Sport and exercise science (Bachelor’s degree)	497	58
Preventive and adapted exercise science (master’s degree)	285	33
Sport science and physical performance (master’s degree)	65	8
Sport management (master’s degree)	7	1
*Participation in non-academic course about exercise-oncology*		
Yes	115	13
No	739	87

**Table 2 ijerph-19-07724-t002:** Students’ knowledge, perception, and confidence to “exercise oncology.”

	All (*n* = 854)			
	Absolutely No	No, Rather Than Yes	Yes, Rather Than No	Absolutely Yes
	%	%	%	%
Do you think that physical activity is important to prevent breast, prostate, and colorectum cancers?	0	4	25	71
Are there teachings, in your degree program, that explain the prescription of physical activity and exercise as a primary prevention of non-communicable diseases?	13	25	26	36
Are there teachings, in your degree program, that explain the physiopathology and treatments of cancer?	22	38	25	15
Are there teachings, in your degree program, that explain the importance of the psychological aspect in cancer?	32	38	19	11
*Is exercise important and effective for psycho-physical health in patients with cancer?*				
Before surgery	1	4	27	69
During anticancer treatments (chemotherapy, radiotherapy etc.)	2	13	32	52
After anticancer treatments	0	1	17	81
*Are there teachings, in your degree program, that explain the prescription of physical activity and exercise for patients with cancer?*				
Before surgery	24	36	23	16
During anticancer treatments (chemotherapy, radiotherapy etc.)	25	38	23	13
After anticancer treatments	23	33	25	19
*Do you have sufficient knowledge and expertise for counselling physical activity to patients with cancer?*				
Before surgery	29	40	23	7
During anticancer treatments (chemotherapy, radiotherapy etc.)	32	43	29	6
After anticancer treatments	28	38	26	9
*Do you have sufficient knowledge and expertise to prescribe exercise in patients with cancer?*				
Before surgery	31	40	21	7
During anticancer treatments (chemotherapy, radiotherapy etc.)	34	43	17	5
After anticancer treatments	30	39	23	8
*Do you have sufficient knowledge and expertise to lead a group-based exercise training for patients with cancer?*				
Before surgery	30	42	22	6
During anticancer treatments (chemotherapy, radiotherapy etc.)	32	46	17	5
After anticancer treatments	28	41	23	7
*Do you have sufficient knowledge and expertise to lead an individual-based exercise training for patients with cancer?*				
Before surgery	29	39	25	8
During anticancer treatments (chemotherapy, radiotherapy etc.)	32	43	19	6
After anticancer treatments	27	39	25	9

## Data Availability

The data presented in this study are available on request from the corresponding author.

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
