# Peer review of "Kinesiology Students’ Perception Regarding Exercise Oncology: A Cross-Sectional Study"

_ijerph, 2022, doi:10.3390/ijerph19137724_

Round 1

Reviewer 1 Report

Basic reporting

The study assessed the knowledge, perception, and competence about exercise in patients with oncological disease in a sample of university students. It is an interesting and well-conducted research topic. Whilst the study undoubtedly has merit, it is necessary to revise a few aspects. 

INTRODUCTION

GENERAL COMMENT: Please follow the journal’s citation style. I am referring to the order of the references.

Despite a good review of the problem, a restructuring of the introduction is needed. Divide the introduction into different sections (paragraphs) in order to help the readers comprehension.

MATERIAL AND METHODS

L 77-89 Indicate the total sample 

Please, could you deeply present how you proceeding to select your sample? 

It is a representative sample? If is it, introduce please the Sample size calculation.

RESULTS

Please, improve table’s format and reduce the total number of tables.

DISCUSSION

L 211: Please, start the discussion with the main purpose and the main findings.

In the discussion, the authors should be more specific to answer the study's aim and address the main results. I find the idea of speculation interesting, but it is important not to shift the focus so far from the results found in the study.

I suggest you to included several considerations about the limitation of your study. For example, there are several variables that you could include as a covariate? Is this a representative sample? 

CONCLUSIONS

The conclusion section should be further developed. 

Moreover, a practical implications section should be included. How these findings can help university statements? Which practical implications can it have on academic curricula?

REFERENCES

Please, follow the journal’s style. 

Author Response

  1. GENERAL COMMENT: Please follow the journal’s citation style. I am referring to the order of the references.

Reply a): Thanks for the observation. We have provided to change the reference style, according to the journal indication.

  1. Despite a good review of the problem, a restructuring of the introduction is needed. Divide the introduction into different sections (paragraphs) in order to help the readers comprehension.

Reply b): Thanks for the suggestion, we have divided the introduction into paragraphs.

MATERIAL AND METHODS

  1. L 77-89 Indicate the total sample 

Reply c). We have modified as requested.

  1. Please, could you deeply present how you proceeding to select your sample? 

Reply d): Thanks for your observation, we have rephrased as follows:

Student eligibility criteria were: i) age ≥18 years, ii) attending a degree in Sport Science, and iii) adequate Italian language proficiency to answer the questionnaire. Potential participants were recruited from the University of Verona, Urbino, Rome ("Foro Italico") and Chieti-Pescara from May to June 2021. Each university invited their students to compete an online survey (Google® Form) through a direct email. Two reminders were sent every two weeks later to increase adherence. A cover letter describing the purpose of the study, the assurance of confidentiality and anonymity was presented before filling the questionnaire.

  1. It is a representative sample? If is it, introduce please the Sample size calculation.

Reply e): thanks for the suggestion. We have added the following in the statistical analysis section.

The minimum sample required was 200 students. The expected sample allowed estimates of binary variables [e.g., percentages of students knowing the exercise guidelines (p) vs. those who do not recognize it (P = 1 − p) with a standard error of 0.035 and a confidence interval between 0.43 and 0.57, assuming the most unfavorable proportion equal to 0.5 (P = 0.5) and alpha 5%.

RESULTS

  1. Please, improve table’s format and reduce the total number of tables.

Reply f): Thanks for the suggestion. We have changed table 2 with graphics as suggested by Reviewer 3, and placed Table 3 and 4 as supplementary materials.

DISCUSSION

  1. L 211: Please, start the discussion with the main purpose and the main findings.

Reply g) Thanks for your suggestion we have added the follows:

To our knowledge, this is the first investigation exploring the knowledge, perception, and confidence towards exercise oncology of students attending four Italian Sports Science Universities. This study highlights that approximately 50% of kinesiologists’ students know the exercise guidelines for patients with cancer, more than 80% recognize the importance of exercise in the cancer continuum, but the majority report the lack of knowledge and expertise for counseling, prescription, and leading of exercise in patients with cancer.

  1. In the discussion, the authors should be more specific to answer the study's aim and address the main results. I find the idea of speculation interesting, but it is important not to shift the focus so far from the results found in the study.

Reply h) Thanks for your observation. The discussion was completely reorganized.

  1. I suggest you to included several considerations about the limitation of your study. For example, there are several variables that you could include as a covariate? Is this a representative sample? 

Reply i) Thanks for your suggestion we have added the following in the limitations section:

Our study presents some limitations. First, the response rate was low, only 13%, suggesting a general interest in the topic, and making our result little generalizable. Secondly, even if the study actively involved 854 university students, the main limitation is related to the involvement of only a limited number of Italian Sports Science Universities. Therefore, the observed results could not be fully representative of the Italian academic situation, also considering the existence of an active national research group, highly contributing to teaching and research activities about the role of PA and exercise in all-grade prevention of cancer. Third, we did not ask in our questionnaire information regarding workplace learning or placements, which may provide important information about students' competence and knowledge. Future studies should consider these aspects to offer a more comprehensive picture of this topic.

  1. The conclusion section should be further developed. Moreover, a practical implications section should be included. How these findings can help university statements? Which practical implications can it have on academic curricula?

Reply j) Thanks for your suggestion we have rephrased as follows:

In conclusion, our study provides an important starting point to improve the training of kinesiologists to be able to stimulate healthcare providers to increase physical exercise prescription and to support patients with oncological diseases with PA counseling, appropriate exercise prescription, and conduction. We found that kinesiology students are aware of the benefits of exercise in the cancer setting. Nevertheless, only a small percentage feel competent to provide PA counseling and program exercise in patients with cancer. Since kinesiologists are one of the experts that may deliver exercise in the cancer population, Sport science universities should implement their academic offer about all-grade prevention of cancer.

  1. Please, follow the journal’s style.

Reply k) Done

Reviewer 2 Report

This is an interesting area and important to check that students have the knowledge and competence to practice in such an area.

Introduction: I think it is important to situate your discussion of the lack of PA engagement for those with cancer in the introduction within the context of low levels of PA in the general population.  eg low levels of engagement in PA is a worldwide issue and contributes to many diseases.

Research question and subsequent conclusion: part of this survey looks at competence but you have not explained why you have looked at such a broad student cohort. It makes sense that students, particularly those early on in their studies, might not feel they have great competence, because that comes about with knowledge, practice and experience. Your analysis does not appear to look at knowledge or competence in relation to years of study or relate to number or duration of work place learning or placements. Surely this will be important to provide some information on. Likewise, the paragraph at the top of page 3 suggests that the bachelors degree is a general preparation course for PA and that it is only one of the masters degrees that provides training for students to adapt exercise in people with chronic diseases. Thus, why did you not just look at those students from the relevant masters degree? Surely that is really the interesting question, do those students feel competent in exercise oncology? Are masters courses meeting that need?

Can students who only complete a bachelors degree qualified to manage clients with chronic diseases? Why did you look at students from all years in these degrees and not just final year students or at least provide a breakdown of their years of studies within your survey? These aspects require addressing in order for the reader to know whether your research, particularly your logistic regression are useful in the real world.

line 139 pg 3 Suggest '78% reported a normal BMI range'. Why have you reported BMI as a characteristic, when we know that there are issues with BMI? If you were looking for an indication of PA level, perhaps whether they met the recommended PA for their age in the past 7 days?

page 4, line 146 suggest 'overall, 44% and 54% of participants identified ...

 page 4, line 148 change 'are' to 'were'

Many of your tables are not very clear for the reader and could be simplified.

For example: Table 2, it took me some time to realise that the line which has No and % on it. No refers to number and not the answer 'no'. Tables could be simplified by providing an n = x to each question and then just the %. Table 2 could be further simplified by just providing the % for the correct answer or at least shading or bolding the correct line for each of those questions.

Table 3 and 4 belong in supplementary material and simpler clearer tables should be added to the main text. 

Author Response

This is an interesting area and important to check that students have the knowledge and competence to practice in such an area.

Reply: Many thanks

  1. Introduction: I think it is important to situate your discussion of the lack of PA engagement for those with cancer in the introduction within the context of low levels of PA in the general population.  eg low levels of engagement in PA is a worldwide issue and contributes to many diseases.

Reply a): many thanks for the suggestion. We have added the following in the introduction:

Most recent worldwide estimates report that approximately a quarter (27.5%) of adults are insufficiently active [1]. Physical inactivity is an established global pandemic that causes an elevated economic burden [2], but it is also a recognized risk factor for premature mortality and several non-communicable diseases, including cancer [3].

  1. Research question and subsequent conclusion: part of this survey looks at competence but you have not explained why you have looked at such a broad student cohort. It makes sense that students, particularly those early on in their studies, might not feel they have great competence, because that comes about with knowledge, practice and experience. Your analysis does not appear to look at knowledge or competence in relation to years of study or relate to number or duration of work place learning or placements. Surely this will be important to provide some information on. Likewise, the paragraph at the top of page 3 suggests that the bachelors degree is a general preparation course for PA and that it is only one of the masters degrees that provides training for students to adapt exercise in people with chronic diseases. Thus, why did you not just look at those students from the relevant masters degree? Surely that is really the interesting question, do those students feel competent in exercise oncology? Are masters courses meeting that need? Can students who only complete a bachelors degree qualified to manage clients with chronic diseases? Why did you look at students from all years in these degrees and not just final year students or at least provide a breakdown of their years of studies within your survey? These aspects require addressing in order for the reader to know whether your research, particularly your logistic regression are useful in the real world.

Reply b) Thanks for this important observation and for giving us the opportunity to clarify this important point. According with the Italian law framework, the master’s degree is mandatory only to teach at school, while also individuals with bachelor’s degree are enabled to program and deliver exercise in patients with cancer. Therefore, we have decided to spread this study to bachelor and master student, in order to identify which students, feel competent in exercise oncology. However, we have added the following in the introduction, in order to better addressing the reader:

Sport science education system in Italy is characterized by two levels of degree: bachelor and master. The bachelor’s degree in Sport and exercise science is a three-year course offering general preparation for PA, including exercise for education, sports, and recovery of physical fitness. After that, three in-depth master’s degrees are currently available: 1) Master’s degree in Sports science and physical performance is focused on sporting and educational activities; 2) Master’s degree in sports management is characterized by marketing and managerial courses related to PA; and 3) Master’s degree in preventive and adapted exercise science aims to train students able to adapt exercise in people with chronic diseases. According to the Italian law framework, also individuals with bachelor’s degree in sport and education science are enabled to program and deliver exercise in patients with cancer.

We agree with the reviewer, regarding the fact that it makes sense that students, particularly those early on in their studies, might not feel they have great competence, because that comes about with knowledge, practice, and experience. We decided to analyze data according to academic degree, because a preliminary analysis considering years of study, particularly 1st, 2nd year of bachelor’s degree vs. 3rd year, does not report significant difference in their response. However, we replaced table 3 and 4 in the supplementary material and we added the analysis that considered 1st, 2nd year vs. 3rd year of bachelor’s degree. Moreover, in the limits section we have added the follows:

Third, we did not ask in our questionnaire information regarding workplace learning or placements, which may provide important information about students' competence and knowledge. Future studies should consider these aspects to offer a more comprehensive picture of this topic.

  1. line 139 pg 3 Suggest '78% reported a normal BMI range'. Why have you reported BMI as a characteristic, when we know that there are issues with BMI? If you were looking for an indication of PA level, perhaps whether they met the recommended PA for their age in the past 7 days?

Reply c) We agree with the reviewer. We have deleted the part regarding BMI.

  1. page 4, line 146 suggest 'overall, 44% and 54% of participants identified ...

Reply d)  Done

  1. page 4, line 148 change 'are' to 'were'

Reply e)  Done

  1. Many of your tables are not very clear for the reader and could be simplified. For example: Table 2, it took me some time to realise that the line which has No and % on it. No refers to number and not the answer 'no'. Tables could be simplified by providing an n = x to each question and then just the %. Table 2 could be further simplified by just providing the % for the correct answer or at least shading or bolding the correct line for each of those questions.

Reply f): Thanks for the suggestion. We have changed table 2 with graphics as suggested by Reviewer 3, and placed Table 3 and 4 as supplementary materials.

  1. Table 3 and 4 belong in supplementary material and simpler clearer tables should be added to the main text. 

Reply g) Thanks for the suggestion, we have simplified Table 3, and added the “old” table 3 and table 4 as supplementary materials.

Reviewer 3 Report

Dear authors,

First of all, thank you for allowing the review of the article that I consider extremely interesting in the field of study that you address.

Regarding the theoretical framework, I consider that it is clear, and concise and that the references are of interest, however, it has a defect in form and that is that the numbering of the references must be done in order of appearance and not in alphabetical order based on the final bibliography. . It is necessary to modify this question to adapt to the regulations of the magazine.

I consider that the work at a methodological level is well designed, however, I do consider that it would have been of interest to complement the study with the contribution of qualitative methodology through a discussion group.

The results are clearly exposed, however, there is excessive use of tables that do not provide additional information to what is described in the text, which is why I consider it of interest to improve fluency and visual load on the reader, provide graphics that clarify the data and make it more fluid. It is an article with too many tables despite the fact that the data and comparisons are of great interest.

With regard to the discussion, it would be necessary to enrich it by contributing a greater number of authors, since I consider that the topic is current enough to provide a richer and more varied discussion.

At the level of conclusions, these are clear and concise; however, it is necessary to add a section where the limitations of this study are explained, as well as a proposal for lines of action that allow researchers to understand where action should be taken to improve and raise awareness about the object of study.

Author Response

Dear authors,

First of all, thank you for allowing the review of the article that I consider extremely interesting in the field of study that you address.

  1. Regarding the theoretical framework, I consider that it is clear, and concise and that the references are of interest, however, it has a defect in form and that is that the numbering of the references must be done in order of appearance and not in alphabetical order based on the final bibliography. It is necessary to modify this question to adapt to the regulations of the magazine.

Reply a) Thanks for the observation. We have provided to change the reference style, according to the journal indication.

  1. I consider that the work at a methodological level is well designed, however, I do consider that it would have been of interest to complement the study with the contribution of qualitative methodology through a discussion group.

Reply b) Thanks for your observation. We have added the contribution of qualitative research as future perspective.

  1. The results are clearly exposed, however, there is excessive use of tables that do not provide additional information to what is described in the text, which is why I consider it of interest to improve fluency and visual load on the reader, provide graphics that clarify the data and make it more fluid. It is an article with too many tables despite the fact that the data and comparisons are of great interest.

Reply c) Thanks for your observation. We have provided to substitute Table 2 with graphics, while Table 3 and 4 have been replaced as supplementary materials.

  1. With regard to the discussion, it would be necessary to enrich it by contributing a greater number of authors, since I consider that the topic is current enough to provide a richer and more varied discussion.

Reply d) Thanks for your observation. The discussion was completely reorganized, inserting other references.

  1. At the level of conclusions, these are clear and concise; however, it is necessary to add a section where the limitations of this study are explained, as well as a proposal for lines of action that allow researchers to understand where action should be taken to improve and raise awareness about the object of study.

Reply e) Thanks for your observation. We have added the following as limitations of the study:

Our study presents some limitations. First, the response rate was low, only 13%, suggesting a general interest in the topic, and making our result little generalizable. Secondly, even if the study actively involved 854 university students, the main limitation is related to the involvement of only a limited number of Italian Sports Science Universities. Therefore, the observed results could not be fully representative of the Italian academic situation, also considering the existence of an active national research group, highly contributing to teaching and research activities about the role of PA and exercise in all-grade prevention of cancer. Third, we did not ask in our questionnaire information regarding workplace learning or placements, which may provide important information about students' competence and knowledge. Future studies should consider these aspects to offer a more comprehensive picture of this topic.

Round 2

Reviewer 2 Report

Suggest - line 85: However, according to the Italian law framework, individuals with bachelor’s degree in sport and education science are allowed to program and deliver exercise in patients with cancer

Table 3: omit % with each figure as each column subheading indicates it is a %.

In the supplementary material. Table 1 and 2: there is no need to have % with each figure as each column subheading indicates it is a %.

Author Response

  1. Suggest - line 85: However, according to the Italian law framework, individuals with bachelor’s degree in sport and education science are allowed to program and deliver exercise in patients with cancer

Reply A): Thanks for your observation. We have rephrased it as you suggest.

  1. Table 3: omit % with each figure as each column subheading indicates it is a %.

Reply B) Done

  1. In the supplementary material. Table 1 and 2: there is no need to have % with each figure as each column subheading indicates it is a %.

Reply C) Done